# MoLD: Fine-Grained Multimodal Risk Assessment via Dynamic Analysis Weights

## Abstract

Current multimodal AI safety detection often lacks granularity, interpretability, and adaptability. To address these limitations, we introduce **MoLD** (Mixture of LoRA Detectors), a framework that uniquely assesses risk by dynamically analyzing the interplay of multiple Low-Rank Adaptation (LoRA) module weights. This approach yields fine-grained, interpretable assessments beyond binary classification, enables concurrent **multi-risk detection**, maintains robustness on long-sequence data, and supports low-cost modularity. Impressively, MoLD demonstrates state-of-the-art (**SOTA**) performance on textual and visual benchmarks while achieving exceptional **few-shot** learning, reducing data requirements by over **90%**. Thus, MoLD provides a powerful, scalable, and data-efficient path to robust, interpretable risk assessment in large-scale multimodal AI systems.

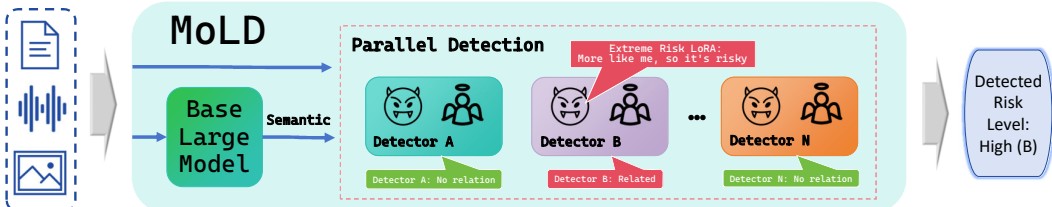

Figure 1: The Teaser Figure of MoLD

## 1 Introduction

While the internet has revolutionized communication and knowledge sharingSingh et al. (2022a), it has also facilitated a surge in harmful online content—spanning text, images, and other media—such as hate speech and harassmentGarg et al. (2023), negatively impacting users and digital environmentsKowalski (2018). Unchecked, these multifaceted issues can escalate into severe social conflictsEzeibe (2021). Consequently, automated content moderation has become a vital research focus. Early efforts, and much subsequent research, concentrated on text security analysis using Natural Language Processing (NLP)Blodgett et al. (2020); Garrido-Muñoz et al. (2021); Weidinger et al. (2021) given the prevalence and complexity of textual data.

Manual content moderation on social platforms struggles with scale, cost, and subjectivity Young (2022); Aroyo et al. (2019). To automate detection, early machine learning approaches evolved from traditional classifiers like SVMs Singh et al. (2023) and CNNs Singh et al. (2022b) to supervised models Burdisso et al. (2019), but these single-task methods lacked multitasking flexibility. Although multi-label classifiers Gunasekara & Nejadgholi (2018); Cai et al. (2024) could address multiple toxicity types, their practical use was limited by the need for complete retraining and large labeled datasets for new categories Zinovyeva et al. (2020).

However, persistent limitations, particularly in handling long-form content Zinovyeva et al. (2020); Caselli et al. (2020) and cohesively assessing diverse online risk modalities, highlight a critical need for a new risk assessment paradigm. Such a paradigm should offer nuanced, interpretable, and data-efficient insights while scaling effectively across various modalities and evolving risks.

To address these limitations, we introduce MoLD, a novel weight-centric framework. MoLD dynamically analyzes multiple groups of LoRA modules, each of which represents a distinct risk theme and contains several LoRAs fine-tuned for different extreme tendencies, in order to uncover subtle risk signals. This enables nuanced, interpretable profiling beyond binary classification, alongside robust few-shot performance. MoLD thus offers a practical, data-efficient tool for large-scale multimodal risk assessment and points towards more adaptable and understandable AI safety mechanisms.

The main contributions of our method are:

1. **Fine-Grained Multimodal Risk Assessment:** The framework introduces a novel, weight-centric approach that analyzes dynamic LoRA weights to provide nuanced and interpretable risk profiles for various content modalities like text and images, moving beyond simple binary classifications.

2. **High Data and Parameter Efficiency:** MoLD demonstrates exceptional few-shot learning capabilities, reducing training data requirements by over 90%. Its modular design allows for the addition of new risk dimensions with negligible parameter overhead, ensuring scalability.

3. **Efficient & Concurrent Multi-Risk Monitoring:** The system is designed to assess multiple, complex risk themes in parallel within a single analytical pass. Its high computational efficiency is maintained across both short-form and long-sequence content, making it ideal for scalable, real-world deployment scenarios.

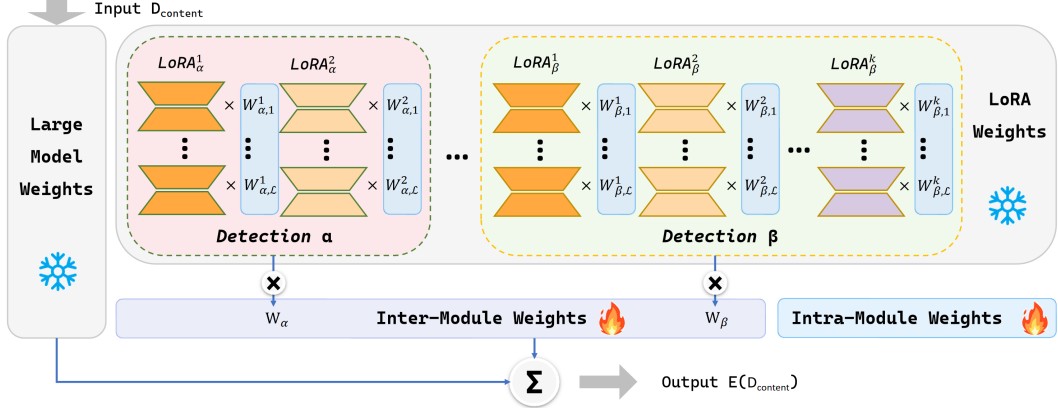

Figure 2: Architecture of MoLD.

## 2 METHODOLOGY

### 2.1 OVERVIEW

This chapter details the MoLD framework, illustrated in Figure 2. At its core, MoLD leverages a base large model for semantic understanding and assesses multimodal content by dynamically optimizing scalar weights for its detection modules. Each detection module is composed of a set of pre-trained LoRA adapters, where each adapter embodies a different extreme tendency of a defined risk theme. The final optimized weights serve as a quantitative measure, indicating both the relevance of each risk theme and the input's specific tendency within those themes.

### 2.2 MOLD ARCHITECTURE

MoLD utilizes a frozen pre-trained base model $(M)$ augmented by N detection modules. Each module $i$ (where $i = 1, 2, ..., N$) employs a set of frozen LoRA $(L_i^1, L_i^2...L_i^k)$, each pre-trained on a different extreme tendency of its theme, thereby focusing the base model's semantic analysis on that specific risk theme.

To assess a given data content, $D_{\text{content}}$ (e.g., an image or text under analysis), MoLD determines the appropriate contribution from each LoRA module by optimizing multiple dynamic scalar weights during the analysis phase:

- **Inter-Module Weights($w_i$):** For each dimension $i$, $w_i$ signifies its overall relevance of that dimension to the assessed $D_{\text{content}}$. These weights are constrained to sum to 1 across all $N$ dimensions (Equation 1), indicating which dimensions are most pertinent to the input.

$$\sum_{i=1}^{N} w_i = 1 \tag{1}$$

- **Layer-Specific Intra-Module Weights($w_{i,l}^k$):** For each layer $l$ in module $i$, the weights $w_{i,l}^k$ quantify the assessed $D_{\text{content}}$'s alignment with each of the module's defined tendencies (indexed by $k$). Constrained to sum to 1 per layer (Equation 2), they measure this layer-specific tendency.

$$\sum_{k=1}^{K_i} w_{i,l}^k = 1, \quad \forall l \in \{1, 2, ..., \mathcal{L}\}, \quad \forall i \in \{1, 2, ..., N\} \tag{2}$$

Where $K_i$ is the number of LoRA adapters within module $i$, and $K \geq 2$, with each adapter representing a distinct tendency, and $\mathcal{L}$ is the total number of LoRA layers.

### 2.2.1 INTEGRATING MULTIPLE DETECTION MODULES

Integrating $N$ detection modules (thus $\sum_{i=1}^{N} K_i$ LoRA modules) with the base model $M$ needs careful synthesis. Naive LoRA combination (e.g., linear weight summation) typically impairs performance and dilutes module specificity. MoLD mitigates this via a hierarchical weighted strategy (Equation 3): Dynamic Intra-Module weights modulate the layer-specific contributions of LoRAs within each thematic group , while Inter-Module weights scale the overall influence of each theme-specific module.

$$E(D_{\text{content}}) = M(D_{\text{content}}) + \sum_{i=1}^{N} \underbrace{w_i}_{\text{Inter-Module}} \cdot \sum_{k=1}^{K_i} \sum_{l=1}^{\mathcal{L}} \underbrace{w_{i,l}^k \cdot L_{i,l}^k(D_{\text{content}})}_{\text{Intra-Module}} \tag{3}$$

### 2.3 LoRA MODULE PRE-TRAINING

The foundation of MoLD's assessment capability lies in its specialized LoRA modules. For each defined risk theme $i$, a corresponding set of $K_i$ LoRA is pre-trained. Each adapter is fine-tuned on a curated dataset representing a specific extreme tendency of that theme.

The rationale for using extreme tendencies is twofold.

1. Using extreme tendencies maximizes the discriminative power of each LoRA adapter and establishes a more expressive semantic basis.

2. By representing novel content as a compositional interpolation of these well-defined extremes, the resulting assessment becomes more fine-grained and interpretable.

This pre-training process is a one-time, offline procedure. Once completed, the weights of the base model ($M$) and all LoRA are frozen. They are not updated during the subsequent risk assessment phase. This clear separation between training and inference ensures that the assessment process is lightweight and fast.

### 2.4 INFERENCE PROCESS

When a new piece of content, $D_{\text{content}}$ (e.g., text or an image), is presented for analysis, MoLD performs a rapid, online optimization process to determine the optimal dynamic scalar weights ($w_i$ and $w_{i,l}^k$).

This process is framed as an inverse problem: what set of weights best configures the frozen model to explain the given content?

The optimization is achieved by minimizing the model's own task loss directly on the content $D_{\text{content}}$. For instance, for a Large Language Model, the objective is to minimize the negative log-likelihood of the content. For an image model, this could be the mean squared error of reconstruction. The loss function is minimized exclusively with respect to the dynamic weights:

$$\min_{w_i, w_{i,l}^k} \text{Loss}(E(D_{\text{content}}), D_{\text{content}})\tag{4}$$

The resulting optimized weights ($w_i$ and $w_{i,l}^k$) form the basis of the final risk classification, as they directly encode the input's alignment with the pre-defined risk themes and tendencies.

## 2.5 ANALYZING WEIGHTS FOR RISK DETECTION

Having established MoLD's architecture and the dynamic weight optimization process, we now detail how these weights are systematically analyzed to perform risk detection. The analysis involves two layers:

1. Determining the relevance of each risk theme via Inter-Module weights.
2. Identifying the specific dominant tendency within the relevant themes via Intra-Module weights.

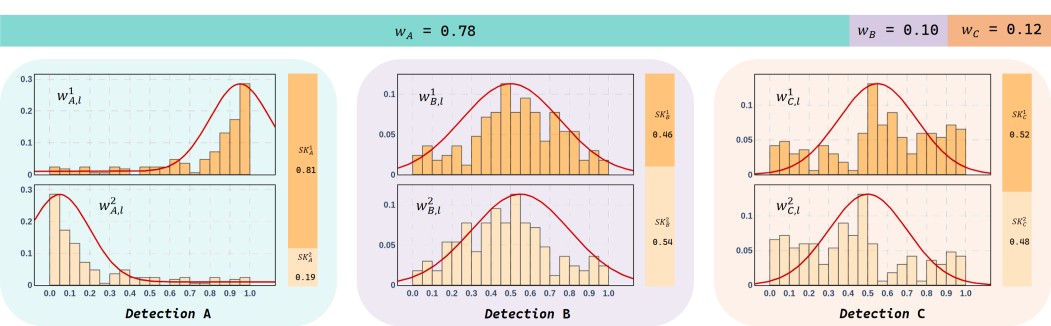

Figure 3: The distribution of weights. Top: Inter-Module Weights, Bottom: Intra-Module Weights Histogram. The histogram is the frequency of the layers of the weight distribution in every 0.05 bin.

### 2.5.1 INTRA-MODULE WEIGHT BIAS ANALYSIS

While Inter-Module weights indicate relevant dimensions, Intra-Module weights show the input's tendency toward an extreme within a dimension i. To quantify this directional bias, MoLD analyzes the distribution of these layer-specific weights as follows:

1. **Intra-Module Weights Distribution:** For a given risk dimension $i$, consider one of its multiple LoRA modules. We analyze the distribution of its layer-specific Intra-Module weights. The $[0, 1]$ range of these Intra-Module weights is divided into discrete bins (e.g., 20 bins, each of width 0.05). The frequency ($f_j$) of weights falling into each bin j is then computed, forming a distribution (Figure 3, bottom) that reveals patterns of weight concentration.

2. **Calculate Modified Skewness** To quantify the directional bias of this distribution, a tailored measure of skewness, $\hat{SK}$, is computed. This metric measures asymmetry relative to the distribution's center (0.5) by assigning a directional sign $x_j$ to each bin's frequency $f_j$. $\hat{SK}$ is calculated as:

$$\hat{SK} = \frac{\frac{1}{n}\sum_{j=1}^{n} x_j \cdot f_j^{\,3}}{\sigma_f^3}, \quad x_j = \begin{cases} -1, & j \leq \lfloor \frac{n}{2} \rfloor \\ 1, & j > \lfloor \frac{n}{2} \rfloor \end{cases}\tag{5}$$

Here, n is the total number, $f_j$ is the frequency of the $j$-th bin, and $\sigma_f$ is the standard deviation of the frequency vector $f = \{f_1, ..., f_n\}$. This metric effectively quantifies the directional activation bias for each LoRA module across layers.

3. **Normalize Tendencies Scores:** For each dimension $i$ and each of its corresponding LoRA modules (indexed by $k$), a modified skewness score, $\hat{SK}_i^k$, is computed. Applying a soft-max function across all these scores for dimension $i$ yields the final Intra-Module risk tendency scores, $SK_i^k$:

$$\{SK_i^k\}_{k=1}^{K_i} = Softmax(\{\hat{SK}_i^k\}_{k=1}^{K_i}) \tag{6}$$

These normalized scores sum to 1, representing the input's alignment strength towards the $k$ extremes of dimension $i$.

### 2.5.2 FINAL RISK CLASSIFICATION

The final risk classification synthesizes dimension relevance (from Inter-Module weights $w_i$) and directional tendency (from Intra-Module scores $SK_i^k$). An input is flagged for a specific risk if both of the following conditions are met for a monitored dimension $i$:

1. **Sufficient Relevance:** The dimension $i$ is identified as highly relevant to the input. This can be determined by its Inter-Module weight being maximal or exceeding a predefined relevance threshold $\theta_w$.

2. **Significant Tendency:** The input demonstrates a strong alignment with one of the dimension's extremes. This is confirmed if the corresponding Intra-Module tendency score ($SK_i^k$) surpasses a sensitivity threshold $\theta_{SK}$.

If these conditions are not satisfied for any monitored dimension, the input is classified as benign concerning these specific risks. The thresholds ($\theta_w$, $\theta_{SK}$) are adjustable to tune the system's sensitivity and specificity for different application needs.

### 2.6 EFFICIENCY AND SCALABILITY ANALYSIS

A key advantage of the MoLD framework lies in its exceptional parameter efficiency and computational scalability, particularly when monitoring multiple risk dimensions concurrently.

### 2.7 PARAMETER EFFICIENCY

Conventional approaches to multi-risk detection often require training and deploying a separate model for each risk type. For a system designed to detect N distinct risk themes, the total parameter count for such a baseline approach ($P_{\text{Baseline}}$) would scale linearly with N:

$$P_{\text{Baseline}}(N) \approx N \times P_{\text{model}} \tag{7}$$

where $P_{model}$ is the number of parameters in a single, fully fine-tuned model.

In stark contrast, MoLD leverages a single, shared base model (M) and only adds a small set of lightweight LoRA adapters for each new risk theme. The total parameter count of the MoLD system ($P_{\text{MoLD}}$) is therefore:

$$P_{\text{MoLD}}(N) = P_{\text{base}} + \sum_{i=1}^{N} \sum_{k=1}^{K_i} P_{L_i^k} \tag{8}$$

where $P_{\text{base}}$ is the parameter count of the large base model, and $\sum_{k=1}^{K_i} P_{L_i^k}$ is the total parameter count of the LoRA for theme $i$. Since LoRA adapters are extremely small by design (typically $< 0.1\%$ of $P_{\text{base}}$), the growth in total parameters as N increases is negligible compared to the baseline.

## 2.8 Computational Efficiency and Parallelism

MoLD's efficiency extends to its inference-time computation. During the risk assessment for a given content $D_{\text{content}}$, the vast majority of the system's parameters (both $P_{\text{base}}$ and all $P_{\text{LoRA}}$) are frozen. The only parameters being optimized ($P_{\text{optim}}$) are the small sets of scalar weights:

$$P_{\text{optim}} = \underbrace{N}_{\text{Inter-Module}} + \underbrace{\sum_{i=1}^{N}(K_i \times \mathcal{L})}_{\text{Intra-Module}} \tag{9}$$

This number is exceptionally small (e.g., in the order of hundreds or a few thousands), allowing the optimization to converge rapidly.

Furthermore, the framework is inherently parallel. The optimization process solves for all $P_{\text{optim}}$ simultaneously, meaning the input content is assessed against all N risk themes concurrently within a single analytical pass. This is a significant advantage over methods that would require N separate models to be loaded and run sequentially. This powerful parallel capability allows MoLD to produce a comprehensive, multi-faceted risk profile with minimal latency, making it an ideal solution for scalable, real-world content moderation systems.

## 3 Experiments

### 3.1 Datasets

To train the LoRA modules representing extremes and evaluate MoLD, datasets for selected textual and visual dimensions were curated and split into training, validation, and testing sets in a 6:2:2 ratio, respectively.

**Textual Datasets:**

We focused on three textual dimensions relevant to societal risks, bias, and style:

- **Gender Perspective:**'Feminism', 'Misogyny'.
- **Racial Attitude:**'Anti-Black', 'Pro-Black'.
- **Behavioral Style:**'Meticulous', 'Careless', 'Humor'.

Data for 'Feminism', 'Misogyny', 'Humor' and 'Anti-Black' categories were extracted from public datasets (ClimaConvoShiwakoti et al. (2024), ToxiGenHartvigsen et al. (2022), DHATEVidgen et al. (2020), and SBICV2Sap et al. (2019)).

Other categories ('Pro-Black', 'Meticulous', 'Careless') were synthetically generated via GPT-4oHurst et al. (2024) using specific prompts (see Table 1 for examples). All textual samples underwent filtering by a GPT-4o assessor (Table 1), retaining only those scoring $> 8/10$ for extremity, yielding 5,000 samples per extreme category.

Table 1: Templates for Large Language Model. Top: Used to generate synthesized data. Bottom: Used to rate the polarity of the content.

| **Prompt Instances** |
| --- |
| • **Prompt for data:** Generate a two-role (Questioner, Responder) dialogue corpus on open-ended/everyday topics. Questioner: Asks/opines curiously, doubtfully, or factually. Responder: Gives serious, goal-driven, detailed, clear, structured, practical, non-vague replies. |
| • **Prompt for score:** As a linguistic analyst, rate conversation meticulousness (Scale: 0=none to 10=extremely meticulous). Consider: Language, tone, context. |

**Visual Datasets:**

For the visual modality, we addressed dimensions related to content safety, and style:

- **Regulated Content:**'Violence', 'Sexual', 'Self-Harm', 'Discriminatory'.
- **Image Style:**'Cartoon', 'Realistic'.

'Sexual', 'Violence', 'Discriminatory' and 'Self-Harm' data were curated from VHD11k Yeh et al. (2024) and UnsafeBench Qu et al. (2024). For 'Cartoon' and 'Realistic' categories, 100 images and 50 short videos per category were collected from public online sources by the authors.

## 3.2 EXPERIMENTAL SETUP

This section details the MoLD risk assessment configuration and the LoRA module training hyper-parameters. All experiments were conducted on a server equipped with 128GB of RAM and a single NVIDIA GeForce RTX 4090 GPU.

### 3.2.1 MoLD RISK ASSESSMENT CONFIGURATION

For the risk assessment procedure detailed, we employed the following specific parameter settings across all experiments:

- **Relevance Identification (Inter-Module Weights):** The most relevant risk dimension $k$ was selected based on the maximum Inter-Module weight ($w_k = \max_i w_i$).
- **Tendency Evaluation (Intra-Module Weights):** A threshold $\theta_{SK} = 0.8$ determined significant directional bias. A value empirically set on a validation set.

**LoRA Module Training Details**

**Base Models:**

- **NLP domain:** Qwen2.5-Instruct series Yang et al. (2024) (0.5B, 1.5B, 3B, 7B parameters)
- **V&L domain:** Pre-trained Stable Diffusion v2.1Rombach et al. (2021).

LoRA modules for each risk theme were trained independently with the following settings:

**NLP LoRAs (Qwen-based):** Learning rate 5e-5 (cosine scheduler), 5 epochs, batch size 8, rank 8, and $\alpha$ as 16.

**Vision LoRAs (SD-based):** Image resolution 512x512, learning rate 1e-5, 400 iterations, batch size 2, rank 4, and $\alpha$ as 0.5.

## 3.3 MoLD ON NLP DOMAIN

This section evaluates MoLD's text-based risk detection via four experiments: few-shot learning, baseline comparison, variable text length handling, and multi-risk detection. Results use Qwen2.5-0.5B-Instruct as the base LLM unless stated otherwise. The dynamic weight optimization for each assessment was conducted over 5 iterations.

### 3.3.1 FEW-SHOT AND CROSS-SCALE PERFORMANCE

MoLD's ability to achieve strong performance with minimal training data was investigated.

**Cross-scale Consistency:**

MoLD showed consistent high accuracy across Qwen2.5-Instruct sizes (0.5B-7B) when trained with few samples (e.g., 200 per extreme). The 0.5B model's few-shot performance matched larger models (Figure 4, top).

**Few-shot Learning vs. Baselines:**

Compared to fine-tuning baselines (BERTDevlin et al. (2019), DistilBERTSanh et al. (2019), RoBERTaLiu et al. (2019), and ClimateBERTWebersinke et al. (2021)), MoLD (0.5B base) achieved

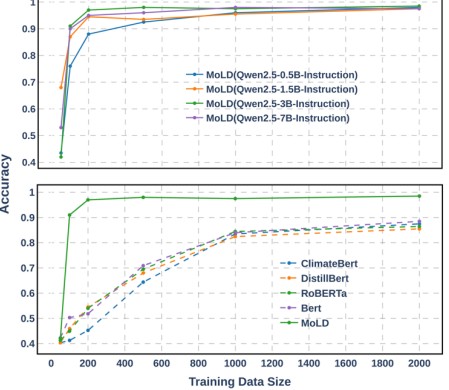

Figure 4: Comparison of MoLD with baselines at different data scales.
Top: MoLD's cross-scale consistency capability.
Bottom: Few-shot learning capability of MoLD

near-peak accuracy ($\sim$ 0.96) with only 200 samples per extreme, while baselines needed $\geq$ 2000 samples (Figure 4, bottom). MoLD required $> 90\%$ less training data. This highlights MoLD's exceptional data efficiency, valuable for low-resource risk detection scenarios.

### 3.3.2 COMPARISON OF BASELINE NLP MODELS

MoLD (trained on 200 samples per extreme) was benchmarked against strong NLP baselines (BERTweet + Llama2Kaya et al. (2024), BERT, DistilBERT, RoBERTa, and Climate-BERT) trained on larger datasets.

MoLD achieved a superior F1 score (0.959), surpassing all baselines (Table 2), demonstrating its competitive performance despite high data efficiency.

Table 3 presents the average inference time over 100 test samples of 30 tokens each. The results show that while MoLD's speed is comparable to the fastest baseline for 2-Risks detection, it holds a significant advantage in more complex multi-Risks (4 and 7) scenarios, demonstrating its excellent scalability for parallel monitoring tasks.

Table 2: NLP Model Performance Comparison.

| Model | Acc | F1 |
|---|---|---|
| BERT | 0.901 | 0.708 |
| DistilBERT | 0.896 | 0.664 |
| RoBERTa | 0.842 | 0.662 |
| ClimateBERT | 0.884 | 0.704 |
| BERTweet+Llama2 | 0.952 | 0.890 |
| Optimal MoLD | **0.96** | **0.959** |

Table 3: Inference Time Comparison.

| Model | 2-Risks | 4-Risks | 7-Risks |
|---|---|---|---|
| BERT | 1882ms | 3766ms | 6580ms |
| DistilBERT | **1027ms** | 2041ms | 3570ms |
| RoBERTa | 1467ms | 2888ms | 5048ms |
| ClimateBERT | 1520ms | 3005ms | 5338ms |
| BERTweet+Llama2 | 1732ms | 3123ms | 6439ms |
| Optimal MoLD | 1150ms | **1546ms** | **1951ms** |

### 3.3.3 DETECTION ACROSS VARIABLE TEXT LENGTHS

MoLD's robustness to varying input lengths was tested by training on short texts (200 samples, 200-400 chars) and evaluating on inputs from 200 to 4,000 characters.

MoLD maintained high accuracy and recall across all lengths (Figure 5), demonstrating its ability to handle long-form content effectively without length-specific adaptations, unlike token-limited models.

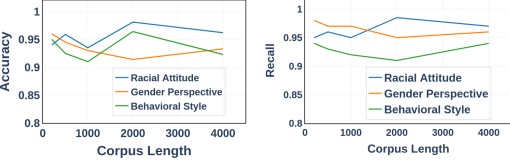

Figure 5: The performance of MoLD in detecting text of different lengths.

### 3.3.4 CONCURRENT MULTI-RISK DETECTION

MoLD's ability to monitor multiple risks simultaneously was tested. Configured with LoRA modules for all three NLP dimensions (Seven extremes) active concurrently.

MoLD maintained high Precision, Recall, and Accuracy across all categories (Table 4), showing minimal interference between parallel detection processes and confirming its scalability for multi-risk monitoring.

Table 4: MoLD performance for multiple text risk dimension simultaneously.

| Risk Dimension | Precision | Recall | Accuracy |
|---|---|---|---|
| Feminism | 0.989 | 0.98 | 0.99 |
| Misogyny | 0.925 | 0.99 | 0.985 |
| Anti-Black | 1.0 | 0.92 | 0.986 |
| Pro-Black | 0.943 | 1.0 | 0.99 |
| Meticulous | 0.923 | 0.96 | 0.98 |
| Careless | 1.0 | 0.92 | 0.986 |
| Humor | 0.956 | 0.934 | 0.976 |

### 3.4 MoLD ON V&L DOMAIN

This section evaluates MoLD's visual risk detection capabilities, comparing it against baselines and testing concurrent multi-risk detection.

### 3.4.1 COMPARISON OF BASELINE V&L MODELS

MoLD's performance on 'Violence' vs. 'Self-Harm' image detection was compared against baselines (LlavaGuardHelff et al. (2024), GPT-4o miniHurst et al. (2024), Gemma3Team et al. (2024) and SG2Zeng et al. (2025)).

MoLD was trained using only 40 images per extreme (tested on 60). MoLD achieved a superior F1 score of 0.901 (Table 5), surpassing reported baseline results on similar tasks. This high performance with minimal training data highlights MoLD's strong data efficiency extending to the visual domain.

Table 5: Comparison of MoLD with baseline V&L models

| Model | Pre | Rec | F1 |
|---|---|---|---|
| LlavaGuard | 0.476 | 0.931 | 0.63 |
| LlavaGuard(SG) | 0.672 | **0.989** | 0.8 |
| GPT-4o mini | 0.683 | 0.977 | 0.803 |
| Gemma3 | 0.777 | 0.879 | 0.825 |
| SG2 | 0.876 | 0.897 | 0.886 |
| Optimal MoLD | **0.896** | 0.91 | **0.901** |

### 3.4.2 CONCURRENT MULTI-RISK DETECTION

MoLD's capacity for simultaneous multi-risk visual detection was evaluated. Configured with LoRA modules for two visual dimensions (six extremes total) active concurrently.

MoLD maintained high performance (accuracy $0.91 \sim 0.967$) across all six categories simultaneously (Table 6). This demonstrates effective concurrent multi-risk visual assessment with minimal interference, confirming its robustness and potential for scalable visual safety monitoring.

Table 6: MoLD detects results for multiple image risk dimensions simultaneously.

| Risk Dimension | Precision | Recall | Accuracy |
|---|---|---|---|
| Violence | 0.94 | 0.922 | 0.953 |
| Sexual | 0.91 | 0.903 | 0.91 |
| Discriminatory | 0.97 | 0.94 | 0.953 |
| Self-Harm | 0.985 | 0.92 | 0.967 |
| Cartoon | 0.99 | 0.93 | 0.96 |
| Realistic | 1.0 | 0.92 | 0.96 |

### 3.4.3 DETECTION ON SEQUENTIAL VISUAL DATA (VIDEO)

To evaluate MoLD's performance on sequential visual data, we tested its ability to detect risks in video clips. Each video was treated as a sequence of image frames. The results (Table 7) show that MoLD maintains high accuracy, successfully identifying the video's dominant. This demonstrates that the framework's robustness to long-sequence data extends from the text to the visual domain.

Table 7: MOLD performance on video risk detection.

| Risk Dimension | Precision | Recall | Accuracy |
|---|---|---|---|
| Cartoon | 1.0 | 0.96 | 0.98 |
| Realistic | 0.98 | 0.96 | 0.97 |

## 3.5 ABLATION STUDY

We conduct ablation studies to validate two key components of MoLD: our modified skewness (SK) analysis and the hierarchical, layer-specific weight architecture. As shown in Table 8, removing either component results in a significant performance drop.

Replacing the SK analysis with a simple averaging heuristic degrades the F1-score from 0.959 to 0.895. The impact is more severe when

Table 8: Ablation studies on core components of MoLD.

| Model Variant | Acc | Pre | Rec | F1 |
|---|---|---|---|---|
| **Full MoLD** | **0.960** | **0.962** | **0.956** | **0.959** |
| w/o SK (Avg.) | 0.915 | 0.906 | 0.887 | 0.895 |
| w/o Layer-Weights | 0.831 | 0.840 | 0.806 | 0.823 |

removing the layer-weights, which lowers the F1-score to 0.823. These results confirm that both our SK-based analysis and the hierarchical weight structure are crucial for MoLD's superior performance.

## 4 CONCLUSION

This work introduced MoLD, a novel framework for nuanced multimodal risk assessment that analyzes the dynamic interplay of specialized LoRA module weights. Experiments demonstrate that MoLD achieves state-of-the-art, scalable, and multi-dimensional risk detection with significant data and parameter efficiency.

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

## A  RELATED WORK

We situate MoLD within existing research by briefly reviewing challenges in multimodal detection, the evolution of automated detection methods, approaches to multi-risk handling, and the novel application of weight-based analysis, thereby highlighting MoLD's contributions.

**Data Scalability in Multimodal Risk Detection.**    A significant challenge in effectively detecting risks across diverse modalities like text and images Lu & Cranefield (2024); Yoon et al. (2025) is the extensive data typically required for training robust models, often involving complex annotation efforts Bai et al. (2024); Van Aken et al. (2018). MoLD directly addresses this data scalability issue through its emphasis on a highly data-efficient learning paradigm.

**Automated Toxicity Detection.**    While advanced architectures such as Pre-trained Language Models (PLMs) like BERT Mazari et al. (2024) and vision models like CLIP Radford et al. (2021) significantly improved contextual understanding and detection accuracy, their effectiveness often relies on large labeled datasets Mahesh (2020). This reliance has spurred the development of more data-efficient techniques, including few-shot Bonagiri et al. (2025); Yeh et al. (2024) and zero-shot learning Cao et al. (2023); AlDahoul et al. (2024). MoLD strategically adopts such data-efficient principles to enhance its practical applicability in resource-constrained scenarios.

**Approaches for Multi-Risk Detection.**    Mixture of Experts (MoE) models Masoudnia & Ebrahimpour (2014); Cai et al. (2024), including LoRA-based variants Chen et al. (2024); Li et al. (2024); Wu et al. (2024), often manage multiple risks by routing or mitigating expert interference Zhou et al. (2022), typically avoiding direct expert conflicts Zhong et al. (2024); Sun et al. (2024). MoLD distinctively analyzes the dynamic weight interplay among multiple thematic groups of LoRA modules, which are intentionally extreme-trained on different tendencies. This assessment through a compositional analysis of these tendencies, rather than expert selection or simple composition Sung et al. (2022); Huang et al. (2023).

**Risk Assessment via Weight Analysis.**    Learned model weights Glorot & Bengio (2010); He et al. (2015) can encode valuable information beyond their primary predictive function Zeiler (2014). While some research explores LoRA weight analysis for auxiliary tasks (e.g., inferring dataset properties Salama et al. (2024); Shokri et al. (2017)), MoLD innovatively applies this concept for direct risk assessment. It analyzes dynamically optimized scalar weights—which modulate the influence of extreme-trained LoRA modules J. et al. (2021), to classify, localize, and score toxicity, forming the core of its interpretable, fine-grained evaluation capabilities.

