# OpenReview forum: "MoLD: Fine-Grained Multimodal Risk Assessment via Dynamic Analysis Weights"
_ICLR.cc/2026/Conference — ICLR 2026 Conference Withdrawn Submission_

### Official Review · Reviewer_wqJx · 2025-10-23

**Soundness:** 3
**Presentation:** 1
**Contribution:** 2
**Rating:** 2
**Confidence:** 3

**Summary:**

This paper introduces MoLD (Mixture of LoRA Detectors), a framework for fine-grained multimodal risk assessment that dynamically analyzes the weights of multiple LoRA modules to identify nuanced risk patterns beyond binary classification.
By optimizing inter- and intra-module weights, MoLD quantifies how textual and visual content aligns with different risk dimensions, enabling interpretable and concurrent multi-risk detection.
Experiments show that MoLD achieves state-of-the-art accuracy in text and image safety detection while requiring over 90% less training data than traditional models.
Overall, MoLD provides a scalable, data-efficient, and interpretable approach to robust multimodal AI safety assessment.

**Strengths:**

+ The idea of using LoRA weight analysis to detect unsafe contents is cool. It is inspired from MoE. It can detect multiple risks together.
+ Performance is good, especially on NLP tasks.

**Weaknesses:**

- Presentation of this paper should be largely improved.
- The set of risk dimensions and their corresponding “extreme” categories appear to be pre-defined and fixed (e.g., three textual dimensions with seven extremes, and two visual dimensions with six extremes). The paper does not clearly justify why these specific dimensions and extremes were chosen, nor does it discuss how the framework would generalize to other or newly emerging risk dimensions. This design choice limits the perceived flexibility and general applicability of the proposed approach.
- The paper relies on GPT-4o–generated synthetic data without human verification. This raises concerns about data authenticity and potential stylistic divergence from real-world online discourse. Consequently, the model may inherit generator-specific biases and its external validity on naturally occurring data remains uncertain.
- The choice of baselines is too limited and not representative of the current state of the field. In the text domain, the evaluation omits frontier LLMs such as GPT-4o, Gemini-2.5, and Claude-4, which are the most relevant comparators for assessing modern few-shot and safety-oriented performance. Similarly, for the visual domain, the study should include stronger multimodal baselines such as GPT-4o, Gemini-2.5, Claude-4, Qwen-VL, and InternVL to provide a fair and comprehensive comparison.
- The theoretical motivation for using skewness as the primary metric to quantify intra-module bias is not sufficiently justified. While the paper empirically shows its effectiveness, it remains unclear why skewness is inherently suitable for representing directional tendencies of LoRA weight distributions. Alternative statistical measures, such as entropy (to capture uncertainty) or kurtosis (to capture tail concentration), might offer comparable or even superior interpretability. The lack of theoretical comparison or ablation across such alternatives weakens the methodological grounding of this design choice.
- The paper does not include an ablation study or sensitivity analysis on the threshold selection (e.g., θ_w and θ_SK), which are critical parameters for determining the system’s detection sensitivity and specificity. Without this analysis, it is difficult to assess how robust MoLD’s performance is to variations in these hyperparameters.
- The paper lacks sufficient transparency and discussion regarding the visual dataset construction. Specifically, as stated in Lines 325–326 (“For ‘Cartoon’ and ‘Realistic’ categories, 100 images and 50 short videos per category were collected from public online sources by the authors”), it is unclear how these samples were selected, whether they introduce potential source or cultural biases, and how such biases might affect the model’s evaluation and generalization. A more detailed description of the data collection process and mitigation strategies for bias would strengthen the work.

**Questions:**

1. Can the MoLD work on multimodal inputs (i.e., having texts and images at the same time)?
2. Is D_{content} text/image or the embedding of text/image?
3. Should discuss the similarity and difference with MoE.

Minor suggestions and typos:
1. Missing spaces before many citations. For example, Line 036: “knowledge sharingSingh et al. (2022a),” Line 038: “hate speech and harassmentGarg et al. (2023).” There are many of that and I just list very few of them. Please proofread the whole paper for the format issues.
2. Missing spaces before left brackets. Line 112: “Inter-Module Weights(w_i),” Line 118: “Layer-Specific Intra-Module Weights(w^k_i,l)”
3. Should use \citep instead of \cite: Line 307-313, Line 325, Line 346-347, Line 377.

---

### Official Review · Reviewer_Fhdu · 2025-10-27

**Soundness:** 2
**Presentation:** 2
**Contribution:** 2
**Rating:** 4
**Confidence:** 2

**Summary:**

This paper presents MoLD, a framework for multimodal risk detection. It freezes a base model and attaches several risk-specific LoRA adapters. During inference, the system optimizes small scalar weights to determine how each adapter contributes to interpreting an input. By analyzing these dynamic weights—particularly using a modified skewness measure—MoLD generates fine-grained and interpretable risk labels. The framework supports text, image, and video inputs, offering efficient, scalable, and concurrent detection of multiple risks.

**Strengths:**

1. MoLD introduces a new perspective on risk detection, which analyzes dynamic LoRA weight behavior rather than model outputs. This weight-centric approach is conceptually original and can generalize across modalities.
2. MoLD achieves strong results with very limited data by reusing a frozen base model and adding small LoRA modules.
3. Unlike traditional models that handle one risk per run, MoLD can analyze multiple risk dimensions in a single inference pass, improving both speed and scalability for real-world tasks.

**Weaknesses:**

1. The evaluation relies mainly on custom and partially synthetic datasets, without additional validation on established benchmarks. While this setup effectively demonstrates MoLD’s internal performance, it leaves some uncertainty about how well the approach would generalize to broader, real-world contexts.
2. The paper claims that combining multiple pre-trained LoRAs with dynamic weighting improves specificity, but it provides no ablation comparing this to training a single multi-task LoRA. Without such evidence, the benefit of the proposed fusion strategy remains unverified.
3. The captions for Figures 1 and 2 are not self-contained, making it difficult to understand the figures without referring back to the main text. Clearer, standalone captions would improve readability and interpretability.

**Questions:**

See weakness

---

### Official Review · Reviewer_cjaf · 2025-10-31

**Soundness:** 2
**Presentation:** 2
**Contribution:** 1
**Rating:** 2
**Confidence:** 3

**Summary:**

The paper proposes MoLD (“Mixture of LoRA Detectors”), which freezes a base model and many LoRA adapters, then optimizes per-input scalar weights across “modules” (risk themes) and their “extreme-tendency” sub-LoRAs during inference to produce fine-grained risk labels.The method integrates LoRAs via a hierarchical weighting, performs on-the-fly weight optimization by minimizing a task loss on the input, and converts the resulting weights into risk decisions using a bespoke skewness-based statistic with thresholds. Claims include SOTA performance, strong few-shot data efficiency (>90% reduction), long-sequence robustness, and parallel multi-risk detection across text, images, and videos.

**Strengths:**

1. A unifying weight-analysis perspective for multi-risk scoring and an attempt to turn LoRA composition into interpretable diagnostics (inter- vs intra-module weights).

2. Clear, modular presentation of the components (integration equation, optimization objective, histogram/skewness analysis) that could inspire future diagnostics work.

**Weaknesses:**

1. Novelty over existing LoRA-composition / MoE literature is unclear. The core mechanism—combining multiple LoRAs with learned mixture weights; here, the novelty is framed as analyzing the learned weights for risk. However, the paper does not rigorously compare against established composition/routing baselines or show why the proposed skewness statistic and two-stage thresholds are preferable to standard discriminative heads trained on features.

2. Evaluation design is fragile and partly synthetic: Several “risk themes” (e.g., Pro-Black, Meticulous, Careless) are synthetically generated by GPT-4o and then filtered by GPT-4o as the assessor, which risks circularity and bias (the same or similar models defining and judging the phenomenon). The paper does not quantify annotator agreement, inter-rater reliability, or distributional realism; nor does it examine failure cases or societal risks from synthetic curation.

3. The paper optimizes per-input weights by minimizing the model’s own task loss on the same content without external labels at inference. It is unclear how this objective is well-posed for risk detection.

4. Most reported metrics are accuracy/F1 (Tables 2, 4–7) with small test sets; there is no emphasis on AUROC/AUPR, FPR@TPR (e.g., FPR95), risk-coverage, or calibration—the standard suite for detection and safety. Safety-oriented multimodal baselines are limited, and it is unclear whether comparisons are conducted on identical data/label protocols.

5. Although claims include long-sequence robustness and parallel risk detection, the paper lacks systematic shift experiments.

6. The proposal of the method is not well-presented or motivated.

**Questions:**

1. Please revise the paper on motivation and inspiration for the proposed method and its novelty beyond standard LoRA and MoE literature.

2. Experimental setup should be refined so as to fully support the claims. For example, using other LLMs to synthesize the dataset.

3. More baselines or appropriate methods should be included, and also some related works can be included.

---

### Note · Authors · 2025-12-05

I have read and agree with the venue's withdrawal policy on behalf of myself and my co-authors.